# Abandoned Intracardiac Electrodes in an Oncological Patient

**DOI:** 10.3390/jpm13060896

**Published:** 2023-05-26

**Authors:** Aneta Klotzka, Sylwia Iwańczyk, Karolina Sobańska, Przemysław Mitkowski, Patrycja Woźniak, Maciej Lesiak

**Affiliations:** Department of Cardiology, Poznan University of Medical Sciences, Długa 1/2 Street, 61-848 Poznan, Poland; sylwia.iwanczyk@usk.poznan.pl (S.I.); patrycja.wozniak@usk.poznan.pl (P.W.); maciej.lesiak@skpp.edu.pl (M.L.)

**Keywords:** electrotherapy, cardiovascular implantable electronic device, sick sinus syndrome, breast cancer

## Abstract

Cardiological and oncological patients comprise the majority of patients admitted to the emergency unit with chronic or acute conditions that are the dominant cause of death worldwide. However, electrotherapy and implantable devices (pacemakers and cardioverters) improve the prognosis of cardiological patients. We present the case report of a patient who, in the past, had a pacemaker implanted due to symptomatic sick sinus syndrome (SSS) without removing the two remaining leads. Echocardiography revealed severe tricuspid valve regurgitation. The tricuspid valve septal cusp was in a restricting position due to the two ventricular leads passing through the valve. A few years later, she was diagnosed with breast cancer. We present a 65-year-old female admitted to the department due to right ventricular failure. The patient manifested symptoms of right heart failure, predominated by ascites and lower extremity edema, despite increasing doses of diuretics. The patient underwent a mastectomy two years ago due to breast cancer and was qualified for thorax radiotherapy. A new pacemaker system was implanted in the right subclavian area as the pacemaker generator occupied the radiotherapy field. In the case of right ventricular lead removal and the need for pacing and resynchronization therapy, guidelines allow a coronary sinus for LV pacing to avoid passing the leads through the tricuspid valve. We facilitated this approach in our patient, suggesting that the percentage of ventricular pacing was very low.

## 1. Introduction

Cardiological and oncological patients comprise the majority of patients admitted to the emergency unit with chronic or acute conditions that are the dominant cause of death worldwide. However, the prognosis of these patients has improved dramatically over the years. Radiotherapy plays a significant role as far as oncological treatment is concerned. Nevertheless, electrotherapy and implantable devices (pacemakers and cardioverters) improve the prognosis of cardiological patients. We present the case report of a patient who, in the past, had a pacemaker implanted due to symptomatic sick sinus syndrome (SSS). A few years later, she was diagnosed with breast cancer.

## 2. Case Report

A 65-year-old female was admitted to the department due to right ventricular failure. The patient has been hospitalized four times for heart failure exacerbations in the past six months. The patient manifested symptoms of right heart failure, predominated by ascites and lower extremity edema. The symptoms occurred despite increasing doses of diuretics.

Six years earlier, the patient had a dual-chamber pacemaker implanted in the left subclavian area due to sick sinus syndrome (SSS). The patient underwent a mastectomy two years ago due to breast cancer. Post-mastectomy, she was qualified for thorax radiotherapy. As the pacemaker generator occupied the radiotherapy field, it was removed two years ago without removing the two remaining leads. A new pacemaker system was implanted into the right subclavian area (Figure 1).

One year after the implantation of a new pacemaker, the patient developed symptoms of heart failure. Within six months, she was hospitalized four times at the local hospital due to ascites and lower limb swelling.

The patient was transferred to our hospital due to another exacerbation of HF. Echocardiography revealed severe tricuspid valve regurgitation (TR) (Figure 2). The tricuspid valve septal cusp was in a restricting position due to the two ventricular leads passing through the valve. The tricuspid valve leaflets had no coaptation. The right ventricle’s function was normal (Figure 3). After radiation therapy, increased scarring was found on the chest skin (Figure 4). One year ago, the patient underwent skin grafts due to healing wounds on her chest. The remaining right ventricle and atrium leads were transvenously removed. However, the procedure did not reduce tricuspid regurgitation (Figure 5).

After one month, during a follow-up hospitalization, the patient was still on diuretic treatment. Symptoms of right ventricular failure were still present, although less severe. Echocardiography still showed tricuspid regurgitation and restrictive positioning of the septal leaflet of the tricuspid valve. Pacemaker control revealed 53% atrial pacing and 6% right ventricular pacing. Due to the small percentage of right ventricle pacing, the remaining functional right ventricular lead was transvenously removed, and a new lead was implanted into the coronary sinus (Figure 6). A pacemaker check confirmed optimal pacing of the atrium and coronary sinus. In a 4-month follow-up, we observed a decrease in tricuspid valve regurgitation, allowing for the discontinuation of diuretics. The patient has not manifested cardiac failure exacerbations over the past six months.

An increasing number of patients with CIED (cardiovascular implantable electronic device) are referred to radiotherapy [1]. Radiotherapy uses high-energy ionizing radiation with X-rays, gamma rays, and charged particles that may cause CIED software and hardware errors, especially if the photon beam energy exceeds 6–10 MV and the radiation dose to which the device is exposed is high (>2–10 Gy). Occasionally, a recommendation is made to transfer the device before starting the radiotherapy, and only if the current position adversely impacts the tumor treatment. This was the matter to tackle when taking care of the presented patient [2,3].

The incidence rate of significant lead-dependent tricuspid dysfunction (LDTD) after CIED implantation is between 10% and 39% [4]. A more significant detrimental impact is that of ICD electrode application and the presence of numerous right ventricular leads. There are still no unanimous guidelines regarding the treatment of tricuspid regurgitation in the presence of CIED leads. General treatment methods include pharmacological treatment involving reducing congestion and removing the lead by reimplanting a new one or using an alternative pacing strategy, such as LV pacing through the coronary sinus or introducing epicardial leads. In the case of our patient, the increasing doses of diuretics did not decrease the number of right ventricular circulatory insufficiency exacerbations. Even though lead-free pacing does not require the use of transvalvular leads, it may also adversely impact the function of the tricuspid valve by influencing the mechanics and causing improper electrical and mechanical ventricle activation [1,5]. On the other hand, surgical repair or replacement of the valve in the case of CIED-induced tricuspid regurgitation is the last resort. The chest scarring could impair median-sternotomy wound healing in our patient’s case.

## 3. Discussion

Thanks to the tremendous progress made in recent years, human life expectancy has increased significantly. This has increased the incidence of cardiovascular disease and cancer and, consequently, increased the number of patients with a cardiac implantable electronic device (CIED) who will require effective oncological treatment, including radiation therapy (RT) [1]. It should be emphasized that both procedures, i.e., the implantation of a pacemaker or cardioverter defibrillator, as well as the use of radiotherapy as part of oncological treatment, have significantly improved the prognosis of cardiac and oncological patients. As the life expectancy of populations increases and oncological and cardiac therapies become more effective, the number of CIED patients requiring radiation therapy for oncological reasons is expected to increase.

A pacemaker is indicated for bradyarrhythmias such as heart block, sick sinus syndrome, atrial fibrillation, and hypertrophic cardiomyopathy. An implantable cardioverter-defibrillator (ICD) is the most effective method for preventing sudden cardiac death due to ventricular fibrillation or ventricular tachycardia. In addition, cardiac resynchronization therapy (CRT) for biventricular pacing was established to treat intraventricular conduction disorder, and an ICD with biventricular pacing (CRT-D) was also developed to prevent sudden cardiac death due to ventricular fibrillation. These implantable devices for treating circulation diseases are called cardiac implantable electronic devices (CIEDs). Radiation therapy uses high-energy ionizing radiation with X-rays, gamma rays, and charged particles, which can cause software and hardware CIED errors, especially if the photon beam energy exceeds 6–10 MV and the radiation dose to which the device is exposed is high (>2–10 Gy). There are two kinds of malfunctions: errors in software and hardware. Software errors include a reset that is changed to a backup set, oversensing that occurs temporarily only during irradiation, and inappropriate ICD operation. Hardware errors cause permanent damage and require replacement. Serious errors are rare and are most often caused by direct irradiation of the device. This can cause irreparable damage to the equipment, requiring the replacement of the device. Soft errors are more common and are related to secondary neutron production by irradiation. Such errors usually involve resetting the device without causing structural damage and can be resolved without replacement.

The risk of CIED malfunction is higher when:photon radiation energy exceeds 6–10 MV (the risk of failure [usually soft errors] is due to secondary neutron production);the cumulative dose reaching the device exceeds 2 Gy (moderate risk) or 10 Gy (high risk)—the dose reaching the pacemaker can be estimated before and measured during the treatment;the patient is dependent on the pacemaker, and the risk of damage to the generator may result in adverse consequences.

Although it is known that factors that increase the risk of adverse RT/CIED interaction include cumulative dose per device (>2 Gy for a pacemaker and >1 Gy for a cardioverter defibrillator (ICD) and beam energy (>6–10 MV), the preservation of normal CIED function during RT is not predictable. Scarce data on the implications of the effect of RT on CIEDs can result in unnecessary disqualification of patients from RT or misclassification of patients for removal of existing CIEDs before RT. Rarely is it an absolute necessity to reposition the device before the start of radiotherapy, and only if the current tumor localization overlaps with the CIED, which will prevent effective RT [2,3]. Our patient had just such a situation, so she had a generator removed before RT, leaving the old electrodes without repositioning. The new pacemaker was implanted with new electrodes on the opposite side. Stratification of the risk of performing radiotherapy in a patient with CIED should be performed by the attending cardiologist/electrophysiologist. It should be based on: CIED location (thoracic vs. external), cumulative dose per CIED and beam energy, and pacemaker dependence or frequent ICD therapy.

Right heart failure (RHF), which occurred in our patient, rarely occurs as an isolated condition. It most often accompanies heart failure secondary to left ventricular failure. We can encounter primary RHF in acute coronary syndromes with right ventricular infarction, arrhythmogenic right ventricular cardiomyopathy, or primary tricuspid regurgitation (TR). We can divide right ventricular failure into acute and chronic. Acute HF accompanies hypotonia or symptoms of cardiogenic shock with low left ventricular filling pressure. In the chronic form, symptoms such as peripheral edema, ascites, features of liver damage, and hepatomegaly are present. In our patient, massive recurrent ascites, peripheral edema, and liver enlargement predominated.

Moderate to severe tricuspid regurgitation is observed in 0.55% of adults. This value increases with age and reaches more than 4% of older people after the age of 75 [6]. In the vast majority (>90%), tricuspid regurgitation is secondary and is responsible for volume or pressure overload of the right ventricle, leading to enlargement of the heart. Secondary regurgitation can also occur due to left heart disease (mitral regurgitation, left ventricular heart failure). Primary causes of tricuspid regurgitation include infectious endocarditis, rheumatic heart disease, carcinoid syndrome, and Ebstein’s anomaly. Electrodes used for electrotherapy contribute to tricuspid regurgitation in 20–30% of patients. These electrodes can cause tricuspid regurgitation by several mechanisms: damage to the valve leaflets, damage to the subvalvular apparatus, or unfavorable interference of the electrode with the valve leaflets, causing a lack of leaflet coaptation. A more detrimental effect is the use of ICD leads and the presence of multiple, including abandoned leads, in the right ventricle. Our patient had two pacemaker leads in the right ventricle, causing significant tricuspid regurgitation [7]. The prognosis for significant tricuspid regurgitation is unfavorable. The assessment of the degree of regurgitation remains a separate issue. Echocardiography remains the examination of choice due to its general availability. In primary tricuspid regurgitation, specific valve abnormalities can be detected. In secondary tricuspid regurgitation, it is necessary to determine the degree of enlargement of the valve annulus, the dimensions of the RV and right atrium, as well as the RV function, as these parameters have prognostic significance. Assessment of RV strain is also an excellent parameter [8]. Due to its high accuracy and reproducibility for RV evaluation, CMR is preferred if this method is available. In the case of our patient, we did not perform CMR, despite the availability of this imaging method at our hospital, due to remaining leads. As the data show, there are no clear recommendations for the management of tricuspid regurgitation associated with CIED. General methods of treating tricuspid regurgitation include pharmacological treatment to reduce conductance and edema. Among diuretics useful in the treatment of right heart failure, those that counteract the activation of the renin-angiotensin-aldosterone system play a unique role. Our patient was taking large doses of spironolactone in addition to furosemide precisely because of her ascites. In addition to pharmacological treatment, it became necessary in our patient’s case to remove the remaining lead by reimplanting a new one or using an alternative stimulation strategy. Such alternative pacing sites include LV access via the coronary sinus, epicardial lead insertion, or implantation of a Micra electrodeless pacemaker. In the patient described, increasing the doses of diuretics did not reduce the number of RHF exacerbations. On the other hand, surgical valve repair or replacement for CIED-induced tricuspid regurgitation appears to be a last resort. In the case of our patient, thoracic scarring could impair the healing of the median sternotomy wound. In clinical practice, tricuspid valve interventions are rarely performed and often too late [9].

Tricuspid valve intervention performed at the right time is crucial to avoiding irreversible RV damage and RHF. The enlargement of the right ventricle or deterioration of its contractility in the course of TR is an indication for surgical intervention. Unfortunately, there are no clear criteria in the guidelines concerning qualifying a patient for surgical treatment of TR. The benefits of surgical repair of isolated secondary tricuspid regurgitation compared to conservative therapy have also not been well established [10].

However, if surgical tricuspid annuloplasty were to be considered for our patient, the recommended procedure would be the removal of the RV lead and the implantation of an epicardial lead. In individual cases, after analyzing the individual benefits and risks, the transvalvular lead can be left in the case of tricuspid annuloplasty. However, it should not be placed between the native and the artificial tricuspid annulus. A mechanical tricuspid valve prosthesis is a contraindication to transvalvular electrode implantation. Percutaneous tricuspid valve repair is a promising method, especially for patients at high perioperative risk. However, in the case of TR associated with CIED, we still need more data regarding this method of treatment.

A large number of intracardiac electrodes can therefore contribute to tricuspid regurgitation. We can divide abandoned electrodes into those that are damaged and those that are still functioning. In the case of electrode dysfunction, electrode abandonment or extraction is possible. The cause of electrode dysfunction can be a broken electrode or damage to the insulation, which causes problems with impedance, detection, or capture of the electrode.

Abandoned electrodes with normal function are most often associated with the device upgrade. Such cases include a pacemaker upgrade to an implantable cardioverter defibrillator (ICD), a change from a dual-chamber system to a single-chamber system with no further indication for the device, or a system transfer for radiation therapy, which was the case in our patient.

Abandoned pacemaker leads or ICDs present clinical challenges. The main issues are (i) clearly defining the risks of leaving an inactive lead and (ii) whether the potential benefits of electrode extraction outweigh the procedure’s risks. There is little data concerning the complications of the remaining leads. Among the complications associated with abandoned leads are problems with venous access and superior vena cava syndrome, system infection, inter-electrode interactions, and tricuspid regurgitation.

In the case of our patient, it was, among other things, abandoned electrodes that contributed to the development of significant tricuspid regurgitation. It emphasizes that the procedure of removing abandoned leads is fraught with greater peri-procedural risk. The “older” the lead, the greater the risk. The management of patients with uninfected leads is still controversial and widely debated [11]. Guidelines do not recommend the extraction of such leads as Class I recommendations, so each decision to remove an electrode must be made by clinicians on a case-by-case basis. Bongiorni showed that removing an electrode “older” than nine years was associated with significantly higher peri-procedural risk. Hence, the most compelling argument the researchers raise in favor of removing abandoned electrodes is the concern that their extraction may become more complex over time. The study also shows that as many as 11% of young patients with abandoned electrodes experience CIED infection, compared to 2% without abandoned electrodes. Electrode migration and electrode-related endocarditis are also complications of abandoned electrodes. Especially in younger patients, the need to remove abandoned leads due to infection should be considered.

There is another indication for abandoned electrode removal: no MRI availability. When the electrodes are connected to a pulse generator, the generator absorbs and dissipates some of the energy. Abandoned transvenous electrodes can be susceptible to heating of the electrode tip by up to about 10 °C. No adverse effects were observed in studies involving patients with abandoned transvenous electrodes who underwent MRI [12,13,14].

The largest of the studies included 80 patients who also underwent chest MRI [13], limited to a specific absorption rate (SAR) < 1.5 W/kg. This study demonstrated the relative safety of a patient with an implanted CIED and abandoned leads, so a 1.5 T MRI (limited to SAR <1.5 W/kg) may be considered in selected patients.

In their research, Polewczyk et al. evaluated 2678 patients undergoing transvenous lead extraction (TLE) in the years 2008–2021. They concluded that the prognosis of patients with LDTVD was worse; however, patients with improved valve function after TLE showed significantly better long-term survival [15]. They reported that TLE with lead reimplantation is a safe and effective option for LDTD management. TLE with omitted tricuspid valve reimplantation can be an alternative treatment option. In cases of ineffectiveness, cardiac surgery with epicardial lead placement should be considered.

## 4. Conclusions

In the case of right ventricular lead removal and the need for pacing and resynchronization therapy, guidelines allow a coronary sinus for LV pacing to avoid passing the leads through the tricuspid valve. We facilitated this approach in our patient, suggesting that the percentage of ventricular pacing was very low.

## Figures and Tables

**Figure 1 jpm-13-00896-f001:**
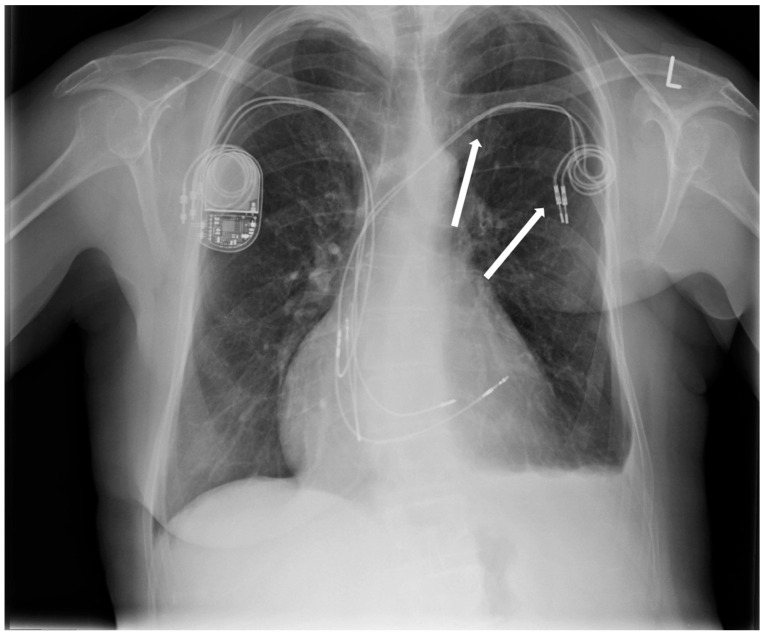
Chest X-ray: abandoned ventricular and atrial electrodes (arrows).

**Figure 2 jpm-13-00896-f002:**
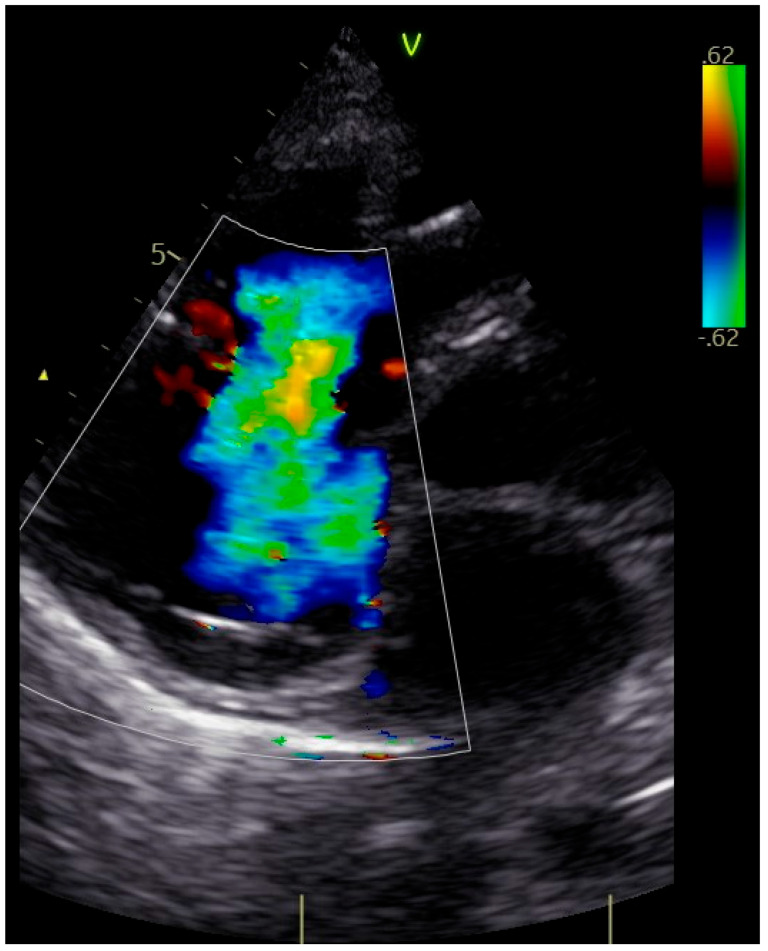
Four-dimensional echocardiographic image—severe functional tricuspid regurgitation (arrow).

**Figure 3 jpm-13-00896-f003:**
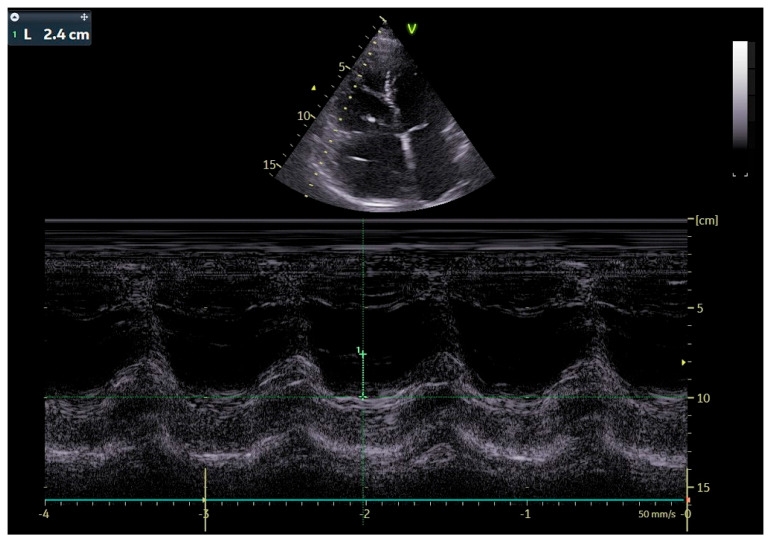
Echocardiographic image, M-mode, evaluation of RV systolic function: tricuspid annular plane systolic excursion (TAPSE).

**Figure 4 jpm-13-00896-f004:**
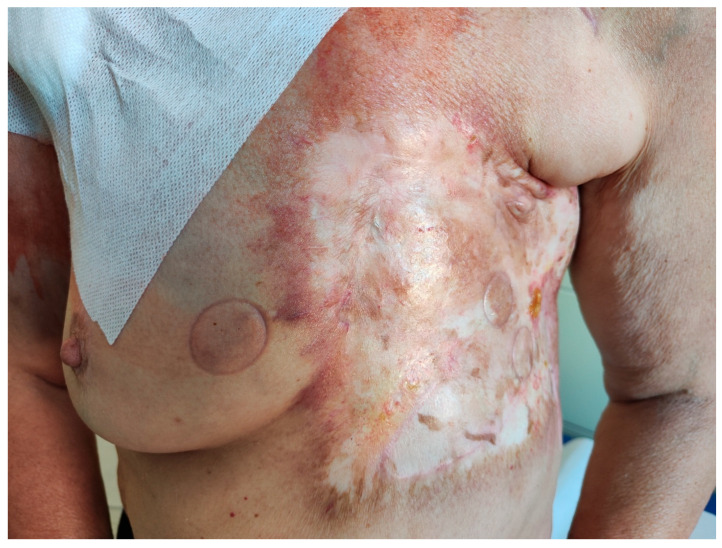
Scarred skin changes after chest radiotherapy.

**Figure 5 jpm-13-00896-f005:**
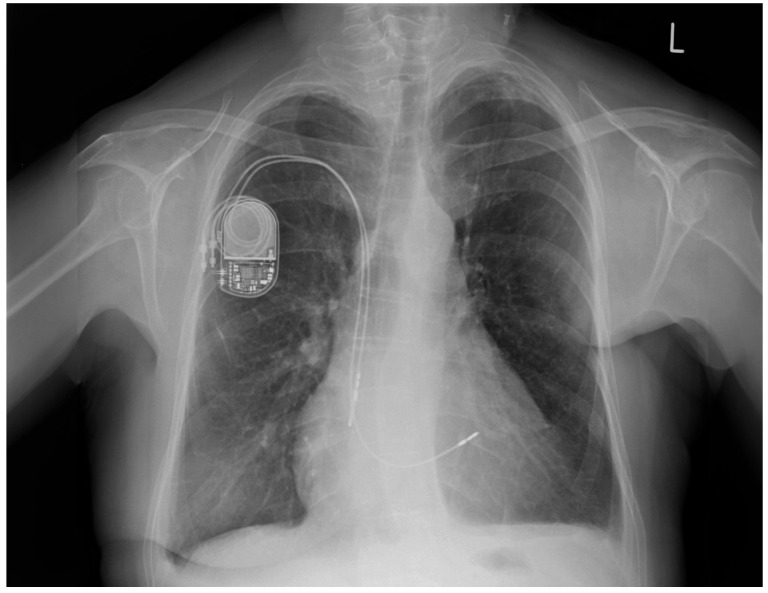
Chest X-ray after removal of abandoned atrial and ventricular electrodes.

**Figure 6 jpm-13-00896-f006:**
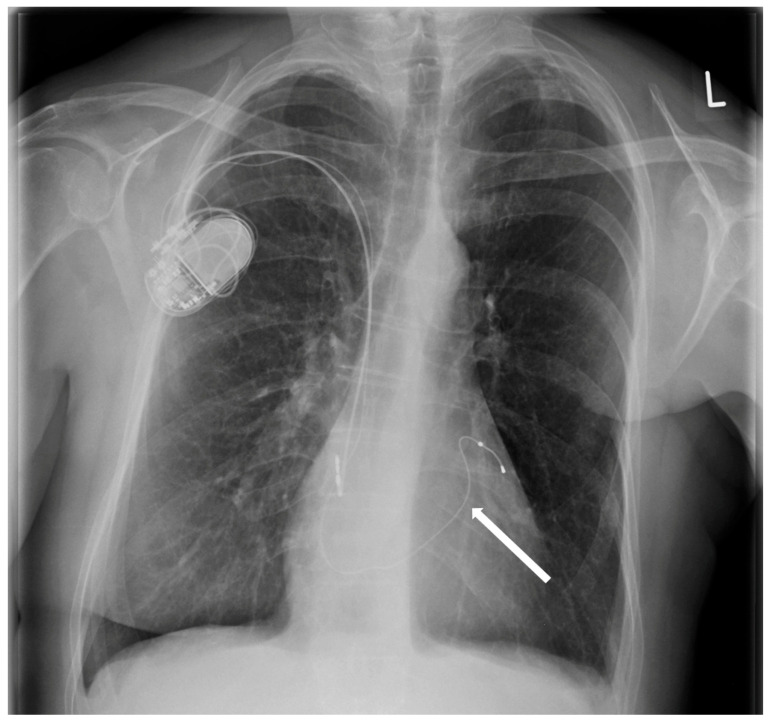
Chest X-ray after removal of the RV electrode and implantation of the electrode into the coronary sinus (arrow).

## Data Availability

The data presented in this study are available on request from the corresponding author. The data are not publicly available due to privacy reasons.

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
