# Peer review of "Abandoned Intracardiac Electrodes in an Oncological Patient"

_jpm, 2023, doi:10.3390/jpm13060896_

Round 1

Reviewer 1 Report

Very interesting informative case report that deserves publication. Two problems occurred in one oncology patient: the presence of abandoned leads at the boundary of the aggressive radiotherapy field and the phenomenon of significant lead-related tricuspid valve dysfunction caused by the passage of one of the ventricular leads through the tricuspid valve. As it turned out, only the removal of the second, later implanted ventricular lead resulted in the release of the tricuspid valve leaflet and the improvement of the function of the tricuspid apparatus. The authors probably omitted the problems of lead extraction from the irradiated field and probably delayed healing of postoperative wounds. The report describes in detail the function of the tricuspid apparatus before the extraction of the leads but omit the most interesting issue - the description of the function of the tricuspid apparatus after extraction of the second ventricular lead. The efficiency of extraction of the lead causing LDTVD is still controversial and therefore completion of the manuscript is highly desirable. In addition, I would suggest citing two papers on the LDTVD issue:

Polewczyk A, Jacheć W, Nowosielecka D, Tomaszewski A, Brzozowski W, Szczęśniak-Stańczyk D, Duda K, Kutarski A. Lead Dependent Tricuspid Valve Dysfunction-Risk Factors, Improvement after Transvenous Lead Extraction and Long-Term Prognosis. J Clin Med. 2021;11:89

Polewczyk A, Kutarski A, Tomaszewski A, Brzozowski W, Czajkowski M, Polewczyk M, Janion M. Lead dependent tricuspid dysfunction: Analysis of the mechanism and management in patients referred for transvenous lead extraction. Cardiol J. 2013;20:402-410

I congratulate the authors on finding a very educational case, proper it management, collecting medical documentation and obtaining the desired treatment effect

Reviewer 2 Report

Dear Authors,

Thank you for giving me the opportunity to read the article. Although it is not a new literature contribution, it is a good article with its reading, fluency and reminders. Therefore, I thank the authors.

I advise authors to reduce the large number of abbreviations in the article. Thus, the reading of the article will be more fluent.

Kind regards.

Minor editing of English language required.
